# Mycobacterial Heat Shock Proteins in Sarcoidosis and Tuberculosis

**DOI:** 10.3390/ijms24065084

**Published:** 2023-03-07

**Authors:** Anna Dubaniewicz

**Affiliations:** Department of Pulmonology, Medical University of Gdansk, 80-210 Gdansk, Poland; aduban@gumed.edu.pl; Tel.: +48-509209455; Fax: +48-585844310

**Keywords:** mycobacterial heat shock proteins, nitrate/nitrite, peroxynitrite, apoptosis, T-cells subsets, Th1/Th2 cytokines, autoimmunity, genetics, sarcoidosis, tuberculosis

## Abstract

Pathological similarities between sarcoidosis (SA) and tuberculosis (TB) suggest the role of mycobacterial antigens in the etiopathogenesis of SA. The Dubaniewicz group revealed that not whole mycobacteria, but Mtb-HSP70, Mtb-HSP 65, and Mtb-HSP16 were detected in the lymph nodes, sera, and precipitated immune complexes in patients with SA and TB. In SA, the Mtb-HSP16 concentration was higher than that of Mtb-HSP70 and that of Mtb-HSP65, whereas in TB, the Mtb-HSP16 level was increased vs. Mtb-HSP70. A high Mtb-HSP16 level, induced by low dose-dependent nitrate/nitrite (NOx), may develop a mycobacterial or propionibacterial genetic dormancy program in SA. In contrast to TB, increased peroxynitrite concentration in supernatants of peripheral blood mononuclear cell cultures treated with Mtb-HSP may explain the low level of NOx detected in SA. In contrast to TB, monocytes in SA were resistant to Mtb-HSP-induced apoptosis, and CD4^+^T cell apoptosis was increased. Mtb-HSP-induced apoptosis of CD8^+^T cells was reduced in all tested groups. In Mtb-HSP-stimulated T cells, lower CD8^+^γδ^+^IL-4^+^T cell frequency with increased TNF-α,IL-6,IL-10 and decreased INF-γ,IL-2,IL-4 production were present in SA, as opposed to an increased presence of CD4^+^γδ^+^TCR cells with increased TNF-α,IL-6 levels in TB, vs. controls. Mtb-HSP modulating the level of co-stimulatory molecules, regulatory cells, apoptosis, clonal deletion, epitope spread, polyclonal activation and molecular mimicry between human and microbial HSPs may also participate in the induction of autoimmunity, considered in SA. In conclusion, in different genetically predisposed hosts, the same antigens, e.g., Mtb-HSP, may induce the development of TB or SA, including an autoimmune response in sarcoidosis.

## 1. Introduction

Sarcoidosis (SA) is a multi-organ disease of unknown etiology. Infectious and non-infectious factors, genetics and autoimmunity have been identified as potential causes of SA (Figure 1) [1,2,3,4,5,6,7,8,9,10,11,12]. The diagnosis of SA is based on a combination of the clinical picture, the radiological picture and, most importantly, the presence of a non-caseating granuloma in the biopsy material of the affected organ. It is necessary to exclude all other possible causes of the development of granulomas, primarily tuberculosis (TB). Since “atypical” TB granulomas without necrosis (non-caseating) can be found in approximately 20% of biopsied TB patients, careful differential diagnosis should be made for SA [3]. Incorrect diagnosis between these two diseases and initiation of immunosuppressive treatment may result in the dissemination of tuberculosis and even the death of the patient. Therefore, SA patients must undergo microbiological and molecular testing for the presence of *Mycobacterium tuberculosis* in the patient’s biological materials (sputum, bronchoalveolar fluid, biopsy material). The interferon-γ blood release assay (IGRA) and/or the purified protein derivative (PPD) skin test are also helpful; however, due to the presence of peripheral anergy in SA, these tests can be false-negative.

Due to the similarities in histopathology between SA and TB, mycobacterial antigens, e.g., *M. tuberculosis* 6 kDa early secretory antigen target (ESAT6), catalase-peroxidase (mKatG), superoxide dismutase A (SODA) and heat shock proteins (Mtb-HSPs) have been recognized as causative agents in the etiopathogenesis of SA [13,14,15,16,17,18].

## 2. Heat Shock Proteins as ‘Danger Signals’

Heat shock proteins, as evolutionarily conserved proteins, are expressed in all pro- and eukaryotic organisms during stress conditions, such as phagocytosis, hypoxia, heat shock, and oxidative stress, as reviewed in [2]. Human heat shock proteins as the main group of ‘danger signals’ (tissue damage-associated molecular patterns-DAMPs) and/or microbial HSP pathogen-associated molecular patterns (PAMPs) recognized by pattern recognition receptors (PRRs) on/in antigen-presenting cells (APCs), may induce autoimmunity in a genetically different predisposed host. Both endogenous DAMPs and exogenous PAMPs via the same PRR often initiate the same signaling cascades and elicit similar responses, especially that many DAMPs and PAMPs can be expressed on host cells as well as on microbes [2,3,4,19,20,21,22,23,24,25].

Abundant amounts of both host and microbial HSP can be generated by low pathogenic bacteria that persist in host monocytes/macrophages for a long time [2,16,25,26,27,28]. Both human and mycobacterial HSP70, HSP65 and HSP16 can be delivered to the APC surface and presented to T lymphocytes in the context of a particular human leukocyte antigen (HLA) and to B lymphocytes, inducing specific cellular and humoral immunity [2,4,5,6,7,8,9,10,11,12,13,14,15,16,16,17,18,19,20,21,22,23,24,25,26,27,28,29,29,30,30,31,32,33,34,35,36,37,38,39,40,41,42,43,44,45,46,47,48,49,50,51,52,53,54,55,56,57,58,59,60,61,62,63,64,65,66,67,68]. Mtb-HSP70, Mtb-HSP65 and Mtb-HSP16 may be the link between infection and autoimmunity [2,31,32,33,34,35,36,37,38].

## 3. *Mycobacterium tuberculosis* Complex and Mycobacterial Heat Shock Proteins in Lymph Nodes from Patients with SA, Patients with TB as Positive Controls, and Patients with Metastatic Non-Small-Cell Lung Cancer and Patients with Nonspecific Lymph Nodes as Negative Controls

In order to check the presence of mycobacteria and mycobacterial HSPs, Dubaniewicz’s team performed a molecular and immunohistochemical analysis of the lymph nodes from patients with pulmonary sarcoidosis [14]. The presence of DNA amplified primers specific for the *M. tuberculosis* IS6110 complex was demonstrated in 3 of 50 (6%) patients with SA. According to the review in 2: [12,13,14,15,16,17,18,39,40,41], Dubaniewicz et al. suggested the involvement not of whole mycobacteria, but of mycobacterial antigens, such as immunogenic Mtb-HSP, in the pathogenesis of sarcoidosis (Figure 2) [14].

Therefore, Dubaniewicz’s team assessed and detected the expression of Mtb-HSP70, of Mtb-HSP65 and of Mtb-HSP16 in all the tested lymph nodes of SA patients and in tuberculous granuloma [14]. In the negative control, weak reactivity of Mtb-HSP70 was found in one patient with non-specific lymphadenopathy. Eight control cases were non-HSP reactive. While in sarcoidosis, the presence of Mtb-HSP70, Mtb-HSPp65 and Mtb-HSP16 was different in the pre-granuloma phase (PSH), granulomas and surrounding lymphocytes. Comparison of all tested mycobacterial HSP expressions between the PSH and granulomas showed a higher reactivity of Mtb-HSP70, Mtb-HSP65 and Mtb-HSP16 in granulomas. The expression of Mtb-HSP16 was much more intense than Mtb-HSP70 in PSH and in the lymphocyte membrane. In sarcoid lymph nodes, the reactivity of Mtb-HSP70 and Mtb-HSP16 was significantly higher than that of Mtb-HSP65. The occurrence of SA in the early stages (I—hilar and mediastinal lymphadenopathy and II—hilar and mediastinal lymphadenopathy with parenchymal changes) revealed that the expression of Mtb-HSP70 and Mtb-HSP16 was significantly higher than that of Mtb-HSP65. Mtb-HSP70 reactivity was significantly more intense in stage II sarcoidosis than in stage I, while Mtb-HSP16 expression was comparable in both stages of SA. Moreover, the reactivity of Mtb-HSP16 was much more intense than that of Mtb-HSP70 in stage I and was comparable to that of Mtb-HSP70 in stage II. The occurrence of Mtb-HSP16 appears to be associated with early-stage of SA, while Mtb-HSP70 is associated with stage II disease. Notably, the reactivity of Mtb-HSP65 was significantly more intense than that of Mtb-HSP70 and Mtb-HSP16 in the capillary wall of lymph node tissues. The reduced expression of Mtb-HSP65 compared to the reactivity of other tested HSPs may also suggest sequestration of the Mtb-HSP65 antigen in a complex form, e.g., in immune complexes (ICs) [14].

## 4. Mycobacterial Heat Shock Proteins in Sera of Patients with SA, Patients with TB and in Healthy Individuals as Controls

Therefore, in the next study, Dubaniewicz et al. [15] assessed the levels of Mtb-HSP70, Mtb-HSP65 and Mtb-HSP16 in the sera of the same group of SA patients, TB patients and an additional control group of healthy individuals (Figure 2). A significantly higher frequency of anti-Mtb-HSP70 antibodies was found in SA and TB compared to the control group. Anti-Mtb-HSP65 and -Mtb-HSP16 antibodies were significantly more present in tuberculosis than in sarcoidosis and healthy individuals. Anti-Mtb-HSP70 antibody levels were comparable in TB vs. SA, while anti-MtbHSP65 and -Mtb-HSP16 antibody levels were similar in all of the tested groups. Significantly higher levels of anti-Mtb-HSP70 antibodies, compared to anti-Mtb-HSP65 and -Mtb-HSP16 antibodies, were detected only in SA. A significantly higher prevalence of anti-Mtb-HSP70 and anti-Mtb-HSP65 antibodies was found in stage II sarcoidosis compared to stage I. These data suggest that Mtb-HSP65 and Mtb-HSP16 may be involved in IC formation and may also be induced in the immune response to SA and TB.

## 5. The Level of Circulating Immune Complexes and the Concentration of Mycobacterial Heat Shock Proteins in Precipitated ICs in the Sera of Patients with SA, Patients with TB and Healthy Individuals

Therefore, in another study [16], Dubaniewicz’s team assessed the level of circulating immune complexes in serum and the concentrations of Mtb-HSP16, Mtb-HSP65, Mtb-HSP70 in precipitated ICs in the same study groups (Figure 2). There was no difference in the incidence between all tested Mtb-HSPs in circulating ICs in patients with active tuberculosis and healthy individuals. In controls and both stages of sarcoidosis, Mtb-HSP16 concentration was significantly higher than Mtb-HSP70 and Mtb-HSP65 levels, whereas in TB only Mtb-HSP16 level was increased compared to Mtb-HSP70. This may explain previously published findings concerning higher serum concentrations of Mtb-HSP70 than Mtb-HSP65 and Mtb-HSP16 or high level of Mtb-HSP16 in lymph nodes in both stages of sarcoidosis in the same group of patients [14,15].

## 6. Concentration of Nitrates/Nitrites (NOx) in the Blood Serum of Patients with SA, Patients with TB and Healthy People

Since Mtb-HSP16 induced by dose-dependent nitrates/nitrites may be involved in latent tuberculosis, active development of tuberculosis or SA, Dubaniewicz et al. [16] assessed serum NOx levels in the same study groups (Figure 3).

Higher concentrations of nitrates/nitrites were found in SA and TB than in controls, but were lower in SA compared to TB. The reduced concentration of NOx can trigger the genetic dormancy program of M. tuberculosis through the higher levels of Mtb-HSP 16 in SA. It seems that Mtb-HSP 16 may be more important than Mtb-HSP70 and Mtb-HSP65 in IC formation and in initiating the immune response in SA associated with the stationary phase of mycobacteria.

Mtb-HSP16 can also protect partially folded proteins during long periods of bacteriostasis, induced by the genetic dormancy program of *M. tuberculosis* [16]. Studies of the genomic and proteomic levels of mycobacteria under stress conditions, especially at low NOx concentration, showed an increased level of the F unit of sigma RNA polymerase (Sig) (formerly katF), which was associated with the accumulation of the Mtb-HSP16 protein in the cell wall during the dormant phase of the bacteria [42,43,44,45,46,47].

There are reports that the *katF* gene, more important than *katG* in turning on the transcription of other genes, produces substances necessary for long-term cell viability [48]. ESAT6, SODA and mKatG, encoded by the *katG* gene, and detected in sarcoid tissues, were also considered in mycobacterial persistence, but less so than Mtb-HSP16, as reviewed in [16,41]. This resistance capacity is observed in all intracellular bacteria. Therefore, this type of adaptation to hostile conditions may be associated with the immunogenic HSP16 *M. tuberculosis* and HSP16 *Propionibacterium acnes*, considered in the etiopathogenesis of sarcoidosis [16,32,42,43,44,45,46,47,48,49,50,51,52,53].

Long-term Mtb-HSP16 stimulation also reduces mononuclear phagocyte apoptosis [16,18,41,49,51,54,55,56,57].

## 7. Peripheral Blood Mononuclear Cell (PBMC) Apoptosis Induced by Mycobacterial Heat Shock Proteins in Sarcoidosis, Tuberculosis and Healthy Controls

Since Mtb-HSP plays an important role in apoptosis, Dubaniewicz et al. [17] assessed spontaneous and Mtb-HSP-induced apoptosis of peripheral blood monocytes, CD4^+^ and CD8^+^ lymphocytes in SA, tuberculosis and controls (Figure 3). The spontaneous apoptosis of monocytes and CD8^+^ T cells was comparable among all of the study groups. Apoptosis of unstimulated CD4^+^T lymphocytes was significantly lower in TB than in controls, and slightly lower compared to SA. Mtb-HSP-induced monocytes and CD4^+^T cell apoptosis were significantly lower in tuberculosis than in SA and in healthy subjects. Mtb-HSP-induced CD8^+^ T cell apoptosis was significantly lower in tuberculosis and sarcoidosis compared to controls. Analysis of PBMC apoptosis before and after stimulation in each study group showed that, unlike those of tuberculosis, sarcoid monocytes were resistant to Mtb-HSP-induced apoptosis, and CD4^+^ T cell apoptosis was increased in SA patients. Apoptosis of CD8^+^ T cells after Mtb-HSP stimulation was significantly increased in all study groups.

It seems likely that dysregulated apoptosis of CD4^+^ T cells and apoptosis-resistant monocytes may be involved in the pathogenesis of SA [17].

The analysis of the apoptosis of monocytes/macrophages has shown conflicting results, as reviewed in [17]. Rutherford et al. [57] found that in SA patients, the products of apoptosis-related genes in PBMCs, in particular the *Bcl-2* family and growth factor genes, were upregulated in a pro-survival profile. Xaus et al. [56], examining the apoptosis of alveolar macrophages (not monocytes) in sarcoidosis, found no differences in pro-apoptotic (*Bax, Bcl-Xs* and *TNFR1*) and anti-apoptotic (*Bcl-2* and *Bcl-XL*) genes. Reduced apoptosis of alveolar macrophages has been demonstrated in TB as a result of increased levels of the anti-apoptotic gene *Bcl-2* [57]. In addition, reduced apoptosis in tuberculosis may be due to a higher production of IL-10 than TNF-α and NO, which is controlled by low expression of the *SLC11A1* (formerly *NRAMP*) gene encoding macrophage protein 1, which is associated with natural resistance [7,51,58,59,60,61,62,63,64,65].

Others [51,65,67] have shown that the virulence of the strain-infecting phagocytes determines not only the production of cytokines, but also the level of HSP and the subsequent process of apoptosis. A high virulence of intracellular bacteria induces the apoptosis of infected monocytes/macrophages, while low virulence inhibits apoptosis and induces the development of an inflammatory process [51,66,67,68]. The high level of HSP produced by a cell infected with a low-pathogenic bacterium inhibits its apoptosis, probably as a result of accelerated maturation of Bcl-2 proteins [68,69]. Previously published data [51,64,66,68,69,70,71,72] suggest an anti-apoptotic effect of mycobacterial HSP70 and HSP65 on mononuclear phagocytes by protecting the integrity of the mitochondrion and/or by combining heat shock proteins with an apoptotic protease-activating factor or the N-terminal fragment of c-Jun, BAG-1 or Bcl-2. In vitro studies have shown that elevated levels of HSP70 reduce the monocyte apoptosis induced by TNF-α or NO and inhibit cytochrome c/d ATP-mediated activation of caspases [32], while Mtb-HSP70 can also induce the release of anti-apoptotic IL-10 [58,59,62,63,64,65].

Highly virulent *M. tuberculosis* induces phagocytes to produce lower amounts of cytokines INF-γ, TNF-α and IL-6, IL-12 than mycobacteria with a low virulence [51,66,67]. This may be the reason for the differences in the production of cytokines and the course of apoptosis in SA and TB obtained by Dubaniewicz et al. [17,18]. In light of these data, the presence of low-pathogenic tuberculosis bacilli in Europe and the USA caused by an increase in mycobacterial infection, and in Japan caused by HSP *Propionibacterium acnes*, may participate in the etiopathogenesis of SA [16].

## 8. Mtb-HSP-Stimulated T Cell Subsets and Th1/Th2 Cytokine Patterns in Peripheral Blood Mononuclear Cell Culture Supernatant from Patients with SA, Patients with TB and Healthy Individuals as Controls

Since the above results revealed differences between peripheral blood mononuclear cell apoptosis in SA and tuberculosis, a subsequent study by Dubaniewicz et al. [17] assessed Mtb-HSP-stimulated T cell subsets and Th1/Th2 cytokine patterns in peripheral blood mononuclear cell culture supernatant from the same study groups (Figure 3). A significantly higher percentage of CD8^+^αβ^+^T cells was found in unstimulated cultures in SA than in controls. Similarly, the concentration of IL-6 was significantly increased and IL-4 decreased in SA, and the concentrations of INF-γ, IL-2, IL-4 and IL-10 were significantly lower in tuberculosis, compared to those in healthy subjects. After Mtb-HSP stimulation, the production of TNF-α, IL-6 and IL-10 increased significantly and the production of INF-γ, IL-2 and IL-4 decreased in SA, while the concentrations of TNF-α andIL-6 increased significantly in TB compared to those the control group. CD8^+^γδ^+^IL-4^+^T cells were detected significantly less frequently in cultures induced with Mtb-HSP in SA than in healthy individuals. CD4^+^γδ^+^TCR lymphocytes were significantly increased in Mtb-HSP-induced cultures in the TB group compared to those in the SA and control groups. Stimulation of Mtb-HSP PBMCs resulted in an increase in the level of pro-inflammatory cytokines, TNF-α and IL-6, in the sera of patients with SA and tuberculosis compared to those of healthy controls. In addition, patients with sarcoidosis showed the lowest levels of IL-4 and the highest of IL-10. The study of the intracellular level of IL-4 in the subpopulations of T lymphocytes showed that CD8^+^γδ^+^IL-4^+^T cells, which are responsible for the reduced concentration of this cytokine in Mtb-HSP-stimulated PBMC supernatants in patients with SA, can induce the development of autoimmune processes.

So far, the involvement of CD4^+^γδ^+^ and CD8^+^γδ^+^ lymphocytes in the immunoreactivity of patients with TB or SA has not been studied, despite existing data in healthy populations.

The predominance of CD8αβ^+^ lymphocytes in SA cells and CD4^+^γδ^+^T cells in tuberculosis may be due to the different response (proliferation and/or cytokine production) of T cells to similar antigens/different epitopes derived from the same *M. tuberculosis* antigens in the context of different HLA haplotypes present in patients with SA or tuberculosis.

According to the results of Dubaniewicz’s group studies, the reason for this disproportion in the content of the subpopulations of T lymphocytes in a group of patients may be the different sensitivity of these cells to non-specific mitogens, as reviewed in [17]. Additionally, the proven direct influence of Mtb-HSP70 and Mtb-HSP65 on the activation of, e.g., CD8^+^T cells may affect lymphocyte disproportion and cytokine production [51,58,59,60,61,62,63,64,69,73,74]. Mtb-HSP70, Mtb-HSP65 or Mtb-HSP16 of any substance αβ^+^TCR and γδ^+^TCR have been shown to produce cytokines TNF-α, IL-6 and IL-10, as well as smaller amounts of INF-γ and IL-2, but not IL-4 [17,35,51,59,62]. Dubaniewicz et al. [17] showed that IL-10 induced by Mtb-HSP inhibits the production of both INF-γ and IL-2, which may be associated with a decrease in CD4^+^T lymphocyte proliferation and a predominance of CD8^+^ T cells, with a reduced ratio of these subpopulations in the peripheral blood of patients with SA [75,76]. Other studies [77] explain the reduced CD4^+^T lymphocyte content and thus the reduced CD4^+^/CD8^+^ ratio by circulating soluble IL-2 (CD25) receptors in both sarcoidosis and tuberculosis.

In addition, reports [78,79] indicate the presence of regulatory cells (CD4^+^CD25brightT and CD4^+^CD25brightFoxP3^+^T) in the blood of SA patients, which may inhibit T-cell proliferation and the production of INF-γ and IL-2 cytokines, but do not inhibit the production of TNF-α. It is known that TNF-α is a key cytokine responsible for sarcoid granuloma formation [1]. Additionally, elevated levels of TNF-α and IL-6 after Mtb-HSP stimulation were detected in patients with SA. The increase in the concentration of these cytokines is characteristic for chronic inflammation, which is present in both sarcoid and tuberculous granuloma formation. Additionally, the increased level of these cytokines may explain the increase in the production of IL-10, a cytokine that inhibits chronic inflammation via negative feedback [80]. Interleukin 10 is produced, e.g., by monocytes/macrophages and Tr1 regulatory cells, which produce large amounts of IL-10 but do not produce IL-4, and this cytokine configuration was observed in SA [17]. Moreover, studies [32,35,80] have shown that Mtb-HSP70 and Mtb-HSP65 stimulate this subpopulation of regulatory cells. In contrast to tuberculosis, Tr1 cells may be responsible for peripheral anergy in SA patients. Different levels of IL-10 and IL-4 after Mtb-HSP stimulation in SA and TB may affect the dominance of the Th2 response and the course of apoptosis of individual subpopulations of T lymphocytes in these two diseases [16,17].

These results indicate that the cause of dysregulation in the proportion of peripheral blood T lymphocyte subsets after stimulation with mycobacterial HSP in patients with SA may also be increased apoptosis of CD4^+^ T cells and reduced apoptosis of lymphocytes CD8^+^, causing a decrease in the ratio of CD4^+^/CD8^+^ lymphocytes in peripheral blood. Among patients with TB, after Mtb-HSP stimulation of PBMC cultures, an increase in the content of CD4^+^ lymphocytes was observed, probably in the course of reduced apoptosis of these cells and/or as a result of the reduced apoptosis of CD8^+^ lymphocytes. Mtb-HSP stimulation of T cells, causing increased apoptosis and thus a lower content of CD4^+^-cells in peripheral blood, as well as reduced apoptosis and a predominance of the CD8^+^ subpopulation in the presence of increased concentrations of TNF-α and IL-10 and decreased levels of IL-2 and INF-γ, may reflect the causes of peripheral anergy encountered in the course of SA.

Previous studies [58,59,60,61,62,63,64] show that the HSP70, HSP65 and HSP16 proteins of *M. tuberculosis* stimulate αβ^+^TCR and γδ^+^TCR lymphocytes to proliferate and produce pro-inflammatory cytokines IL-1β, IL-6, IL-8, IL-12 or TNF-α and anti-apoptotic IL-10, TGF-β or NOx. According to the authors [59,62,65], Mtb-HSP65 protein activates not only Th1 lymphocytes, but also monocytes/macrophages to produce IL-1β and TNF-α. Additionally, other in vitro studies [63] have shown that stimulation of Mtb-HSP T cells, especially HSP70, induces increased production of anti-apoptotic IL-10, which inhibits the production of INF-γ and IL-2, as reviewed in [16,18].

Reducing the apoptosis of T lymphocytes and monocytes/macrophages in the presence of intracellular persistent antigens may lead to the development of an inflammatory process with the constant presence of pro-inflammatory cytokines [81]. One of the possible mechanisms of the pro-inflammatory effect of these cytokines may be the regulation of NO production. Studies by other authors have shown that cytokines INF-γ and TNF-α increase the level of NO synthase and can induce programmed cell death, while cytokines IL-4, IL-10 and TGF-β, which inhibit NO production, can reduce the process of apoptosis of these cells [51,55,81]. Chronic stimulation of Mtb-HSP16 also downregulates inducible nitric oxide synthase with low nitric oxide and mononuclear phagocyte bactericidal activity, followed by persistent antigenemia, as reviewed in [16].

## 9. Concentration of Peroxynitrite (ONOO^−^) in Supernatants of PBMC Cultures Treated with Mtb-HSP in Patients with SA, Patients with TB and Healthy Individuals

Since NOx reduction can also result from the reaction of NOx with peroxide and the subsequent production of ONOO^−^, Dubaniewicz et al. [54] assessed peroxynitrite levels in supernatants of Mtb-HSP-induced PBMC cultures from the same research groups (Figure 3). Significantly higher concentrations of ONOO^−^ with Mtb-HSP stimulation were found in patients with SA and tuberculosis than in the control group. However, significantly higher levels of ONOO^−^ after Mtb-HSP induction were found in sarcoidosis than in tuberculosis. An elevated concentration of peroxynitrite may explain the low level of NOx, and due to the release of cryptogenic epitopes of autoantigens, it may induce autoimmunity, considered in sarcoidosis [33,34,35,36,37,38,54,82,83].

## 10. Microbial Heat Shock Proteins and Autoimmunity in Sarcoidosis

Mycobacterial HSP70 has been reported to delay the maturation of bone marrow-derived murine APCs, which are known to not only drive differentiation into Th1 or Th2 profiles, but also to induce T cell tolerance or activation of CD4^+^CD25^+^ regulatory T cells [37,63,84,85,86]. Mycobacterial HSP70 or HSP65 modulating the expression of co-stimulatory and adhesion molecules, e.g., B7, BTNL2, CD80/86 or ICAM-1, may participate in the initiation and maintenance of autoimmune responses [2,12,85,87]. In addition to an abnormal level of regulatory cells and the expression of costimulatory molecules, clonal deletion, epitope spreading, polyclonal activation, bystander activation and molecular mimicry between human and bacterial HSPs are also responsible for the loss of self-tolerance and the development of autoimmunity [2,12,19,20,34,37,85,87,88,89,90,91,92,93,94,95].

Heat shock proteins, as the target of humoral and T cell-mediated immune responses to infections, may be a link between the immune response to infection and autoimmunity induced by T cell cross-reactivity, not only between bacterial and human HSPs, but also between pathogens [2,12,33,85,95]. Moreover, a γδ^+^T cell clone reactive to Mtb-HSP also responds to homologous human HSP, suggesting a possible link between infection and autoimmunity [96]. Moreover, since the closely related phylogenetically genera *Mycobacterium, Corynebacterium* and *Streptomyces* share an extended gene set with *Propionibacterium*, the homology between bacterial HSPs is greater, with 78% identity between Propionibacterium and Mtb-HSP60, and 67% identity for Mtb-HSP70 [42,53,97,98,99]. It is worth noting that HSP70, HSP65 and HSP16 of *M. tuberculosis* or *M. lepra* are identical to the HSPs of *M. bovis* BCG (bacillus Calmette-Guerin) [42,99]. In addition, the mycobacteria HSP16, HSP65 and HSP70 share 18–60% identity with their human homologues [[12], reviewed in [16],[97]]. A study by Esaguy et al. [91] showed that *M. leprae* and *M. tuberculosis* HSP65 is specifically recognized by antibodies directed against the human lactoferrin (Lf) protein. They suggested that cross-reactivity of the antigenic protein Lf-mycobacterium HSP65 may contribute to the formation of autoantibodies/immune complexes in autoimmune diseases. Identification of HLA class I and II restricted T cell epitopes from host proteins showed that they are similar to *M. tuberculosis* antigens [95].

Epitopes, especially KPLVIIAEDVDGEALSTLVLN, mycobacterial and homologous human HSP60 are associated with multiple alleles including HLA-DRB1:*0101,*0301,*0401,*0701, *0802,*1101,*1501, with high affinity in both multiple sclerosis and rheumatoid arthritis [95]. Individual epitopes of mycobacterial HSP70, HSP65 and HSP16 peptides that selectively bind to HLA-DR1, DR2, DR3, DR4, DR7, DR53, DQB1*0302 and DQA1*0301 affect the regulation of the body’s immunoreactivity to *M. tuberculosis* infection. Some of these HLA antigens, e.g., DR3 alleles, DQB1*0302, DQA1*0301, DQB1*0201 and DQA1*0501 may be associated with a high risk of developing autoimmune diseases (reviewed in [6]). Structures of HLA-DR and HLA-DQ molecules with bound peptides from candidate (auto)antigens have shown that critical polymorphic residues (pockets) determine the interaction of these HLA molecules with peptides in sarcoidosis, mycobacterial infections and autoimmune disorders (reviewed in [6]). Presentation of HLA-restricted peptides by *M. tuberculosis*-infected macrophages may be of importance for Asian tuberculosis patients expressing a specific P9 pocket on DQB1*0503 alleles. On the other hand, susceptibility to SA has been associated with pocket 7 on DRB1*1101,*1201,*1501 in Americans. The amino acid residue on DQB1*0602 likely affects pocket 4 in Americans, and the amino acid residue on DRB1*01,*04 alleles may affect pocket 6 in Europeans, which may be important for protection against SA. Peptide binding studies have shown that the amino acid residue on DQB1*0602, which likely affects pocket 4, has a significant effect on the repertoire of self-peptides that can be presented by these HLA class II molecules, especially the haplotype A1/B8/DR3/DQ2(DQB1*0201/DQA1*0501)/DQ8(DQB1*0302/DQA1*0301), as reviewed in [6].

Mycobacterial HSPs, e.g., HSP65 or HSP70, can also activate macrophages and dendritic cells for the release of pro-inflammatory cytokines and nitric oxide via toll-like receptors (TLR), especially TLR4 [19]. Abnormal TLR, especially TLR2, TLR4 and TLR9 activation can contribute to sarcoidosis and other autoimmune diseases [24,25,100,101]. TLR9, in which polymorphism was detected in some chronic diseases, recognizes, e.g., fragments of mycobacterial wall or DNA, bacterial and human amyloids/DNA complex, antigens of *Propionibacterium acnes*, as well as chemicals and heavy metals. Bacterial amyloids are molecules that the immune system recognizes as a conserved molecular pattern (PAMPs), and they may be the link between infections and autoimmunity. Furthermore, amyloids are also expressed by humans as beta amyloid and serum amyloid A (SAA). They have high structural and functional similarity to bacterial amyloids. Thus, amyloids may mimic host molecules in order to hide from the immune system. In return, the immune system recognizes these molecules regardless of their origin, which may initiate autoimmune responses in genetically predisposed individuals. Serum amyloid A may have the potential to form poorly degraded amyloid b (Ab), which may play a role in sarcoid granuloma development. Amyloid b, as a DAMP, can induce chronic inflammation via receptors PRRs and the receptor for advanced glycation end products, as reviewed in 2, [102,103,104]. SAA, as an abnormal product of polymorphic *SAA1-3* genes, induces a prolonged production of DAMPs and heat shock proteins, especially HSP16 and HSP70. Heat shock protein 16 reduces the cytotoxicity of the Ab peptide by binding to it, resulting in an alteration of its conformation, as reviewed in 2, [103].

Moreover, in the pathogenesis of infectious and autoimmunological disorders, during ‘NETosis’, the third programmed neutrophil cell death, neutrophil extracellular traps (NETs) are formed following the tissue damage and the consequent release of DAMPs e.g., HSPs, as reviewed in [105,106]. On the other hand, under the circumstances of DAMPs such as HSPs, HMGB1, RNA and DNA of host origin, as well as PAMPs such as bacterial, fungal, or parasitic infection, microbial components (lipopolysaccharide and lipoteichoic acid) and oxidative stress are detected as the initiators of NETosis [107]. Heat shock protein 72 has been found in apoptotic and necrotic cells and in DNA within NETs, where it is also necessary for the elimination of *M. tuberculosis* [108,109].

In conclusion, the response to Mtb-HSP differs among SA, tuberculosis and healthy individuals, possibly due to different genetic backgrounds (Figure 3).

## 11. Genetic Predispositions (HLA Class I and II, *SLC11A1*, *FCGR*) of Patients with SA, Patients with TB and Healthy Controls (Figure 3)

Dubaniewicz et al. [4,6] evaluated associations of HLA class I and II alleles with SA and tuberculosis in a homogeneous Caucasian group. In most cases, SA and TB had the opposite frequency of HLA class II alleles.

Among Polish patients with SA, antigens HLA-B51(5) and HLA-B8 were more frequent, and antigens HLA-B13, -B35 and -Cw4 less present than in the control group. However, after Bonferroni correction, only the HLA-B35 antigen was found to be significantly different in SA patients and controls. In tuberculosis patients, HLA-B62(15) and HLA-Cw5 antigens were more common and HLA-A2 less frequent compared to the control group, but only differences in B62(15) and Cw5 were significant after Bonferroni correction. HLA-B51(5) and HLA-B8 antigens were more frequent and B13, B62(15), Cw4 less present in SA than in tuberculosis and remained significant after Bonferroni correction. The frequency of other tested antigens in both populations was comparable [4].

In SA, DRB1:*03,*11, DQB1*02 and DQA*0501 alleles were less frequent in Stage I with Löfgren syndrome (acute disease) and DRB1*15 and DQA1:*0102,*0103 in Stage II alleles, while DRB1:*16,*04,*08, DQB1:*03,*04,*05,*06 and DQA1:*0102,*0301 alleles were less frequent in both stages of SA than in the control group. After Bonferroni correction, only DRB1*04, DQB1:*02,*03,*05,*06 and DQA1:*0102,*0301,*0501 alleles differed significantly. In tuberculosis, DRB1:*16,*14, DQB1*05 and DQA1*0303 were more frequent, and DRB1*11, DQB1*02 and DQA1:*0201,*0505 less frequent compared to the control group. However, after adjustment, the incidences of only DRB1*16, DQB1:*02,*05 and DQA1:*0303,*0505 were significantly different.

After correction, the DRB1*11 allele was more common and DRB1:*16,*04,*14, DQB1:*03,*05 and DQA1:*0301,0302,*0303 alleles less frequent in both Stages of SA than in tuberculosis. DQB1*02 and DQA1:*0201,*0501 alleles in Stage I and DRB1:*15,*13 in Stage II were more common in SA than tuberculosis, but after correction, only DRB1*15, DQB1*02 and DQA1*0501 significantly differed. The DQB1*02 and DQA1:*0201,*0501 alleles were more frequent only in acute SA, while the allelesDRB1:*15,*13 were more common only in Stage II SA compared to tuberculosis [6].

These results, in agreement with most studies, indicate an increased frequency of the DRB1*03, DQA1*0501 and DQB1*02 alleles, characteristic of autoimmune diseases, in Stage I Löfgren’s syndrome, while the presence of DRB1*15 was more frequent in Stage II SA. Contrary to other reports, DRB1*11 was positively correlated with a high risk of developing acute SA.

Increased presence of DRB1:*16,*14 and DQB1*05 alleles have also been reported by other in TB patients. A positive association with DQB1*05 has also been revealed in SA patients from other ethnicities [reviewed in 6]. A correlation between DRB1:*11,*15 alleles and the presence of *M. tuberculosis* DNA was detected in chronic SA, while the presence of DRB1:*03,*04 alleles was associated with the absence of mycobacterial DNA in acute SA [110].

It should be noted that the allele frequencies of DRB1*16, DRB1*11, DQB1*05 and DQB1*02 in tuberculosis were approximately opposite to the presence of these alleles in both Stages of SA compared to controls. These results obtained by Dubaniewicz et al. [6] suggest that the DRB1*16 and DQB1*05 alleles may be associated with an increased susceptibility to TB and may decrease susceptibility to SA. On the other hand, the association of the frequency of DRB1*11 and DQB1*02 alleles with a low risk of developing tuberculosis may favor the development of SA. 

The specific expression patterns of DR and DQ alleles appear to be adversely correlated in SA and TB. In addition, a study of non-HLA genes by Maliarik et al. [111] showed that *SLC11A1* polymorphisms associated with an increased susceptibility to TB may also be involved in SA protection. In the same groups tested, the 3 allele in the functional promoter region (GT)n, repeated in the *SLC11A1* polymorphism, was significantly associated with SA compared with tuberculosis and healthy controls [7]. Additionally, a significant decrease in the presence of the 2 allele associated with tuberculosis in many populations and an increase in the presence of the 3 allele, consistent with previous research on autoimmune diseases, may suggest a link with the susceptibility to developing SA, but may also be involved in protection against tuberculosis [7,111].

Previous functional studies have shown that the 3 allele in the promoter polymorphism (GT)n drives high levels of reporter gene activity, even in the absence of exogenous stimuli. By implication, this allele would therefore be associated with a high expression of the NRAMP1 protein, a high activation status of macrophages, e.g., HLA or NO, and an enhancement of pro-inflammatory responses, including TNF-α, IL-1β and chemokine responses.

In another molecular study, Dubaniewicz et al. [8] assessed the polymorphism of the *FCGR* genes encoding the receptor for the Fc fragment of immunoglobulin G IIa, IIb, IIc, IIIa and IIIb (FcγRIIa, FcγRIIb, FcγRIIc, FcγRIIIa and FcγRIIIb, respectively) plays a role in enhancing the circulation of immune complexes with the presence of M heat shock proteins in the case of tuberculosis in patients with sarcoidosis. The polymorphism of the *FCGR2A*, *FCGR2C* and *FCGR3A* genes found in patients with sarcoidosis may cause dysfunction of the Fcγ receptors encoded by them, and thus explain the previously observed increased immunocomplexemia with a simultaneous increase in the percentage of FcγR monocytes in the same patients [16,112,113]. In addition, the differences found in the percentage of individual alleles and genotypes of the *FCGR2A* and *FCGR2C* genes may suggest the presence of a different pathomechanism of the development of Stages I/II and III/IV SA, and explain the previously demonstrated greater immunocomplexemia in Stage I/II than in sarcoidosis in Stage III/IV [16,113]. However, differences in the frequency of individual alleles and genotypes of the *FCGR3A* gene may indicate the existence of a different pathomechanism of the development of sarcoidosis in Stages I and II, with particular emphasis on the possible involvement of the autoimmune process in the first stage of the disease. In contrast to Stage I sarcoidosis, there were no differences in the presence of polymorphic *FCGR3A* gene variants in patients with Stage II SA and patients with tuberculosis, which may suggest a common etiopathogenetic factor of chronic sarcoidosis and tuberculosis.

In conclusion, in different genetically predisposed hosts, the same antigens, e.g., Mtb-HSP, may induce the development of TB or SA, including an autoimmune response (Figure 3).

## Figures and Tables

**Figure 1 ijms-24-05084-f001:**
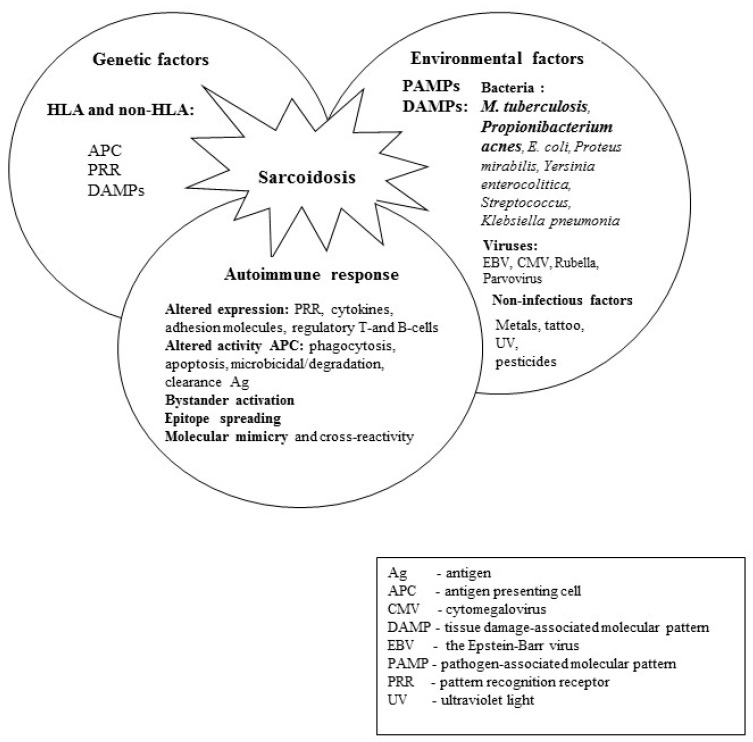
The etiopathogenesis of sarcoidosis.

**Figure 2 ijms-24-05084-f002:**
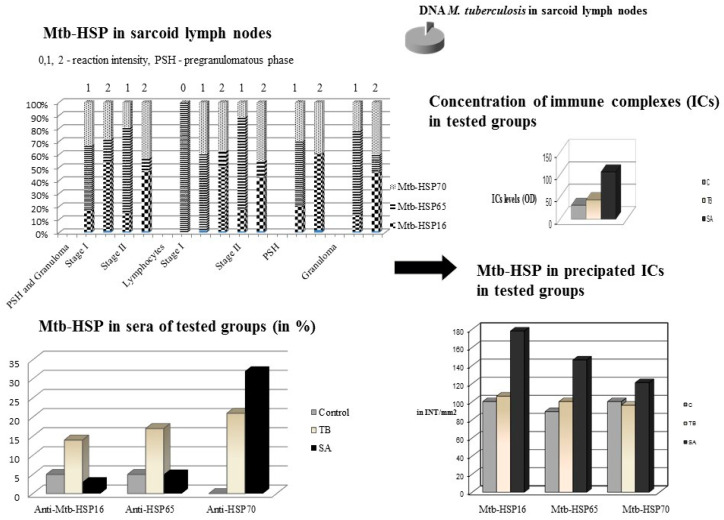
The presence of mycobacterial heat shock proteins in lymph nodes, sera, and in precipitated immune complexes in patients with SA, patients with TB and the controls.

**Figure 3 ijms-24-05084-f003:**
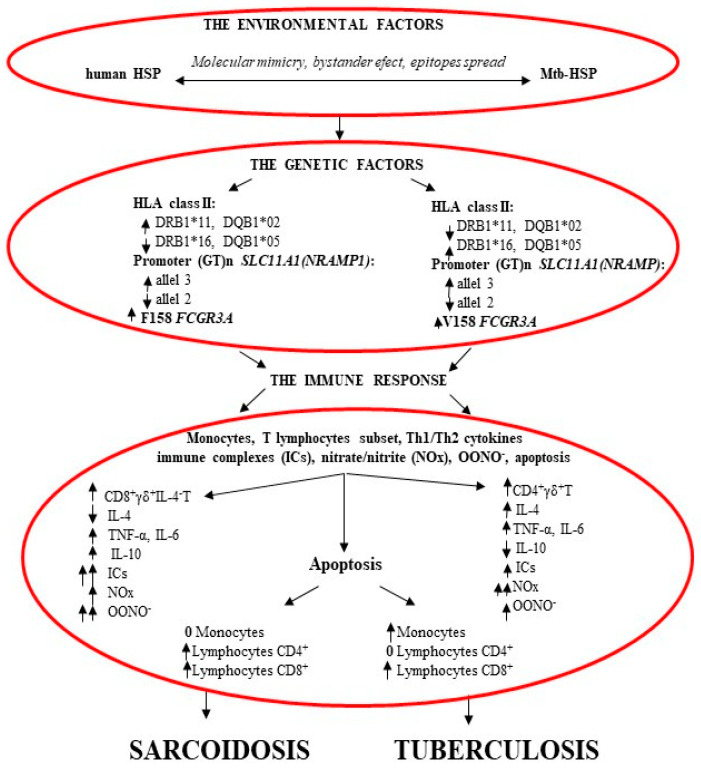
Different roles of mycobacterial Heat Shock Proteins in the pathogenesis of sarcoidosis and tuberculosis.

## Data Availability

The data presented in this study are available on request from the corresponding author.

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
