# Peer review of "Mycobacterial Heat Shock Proteins in Sarcoidosis and Tuberculosis"

_ijms, 2023, doi:10.3390/ijms24065084_

Round 1

Reviewer 1 Report

In this article Author review the role of Mtb Heat Shock Proteins in the pathogenesis of Sarcoidoisis and Tuberculosis. The manuscript is well written and following are some comments and concern. 

1. In the introduction section author should detail the major differences between Sarcoidosis and Tuberculosis both from pathogenesis and diagnostic perspective. 

2. There are studies indicating the susceptibility toward Sarcoidosis which the author should highlight in this review. 

3. Author should provide a pictorial depiction of differential HSP roles in the pathogenesis of Sarcoidosis and Tb.

4. For easy reading author should tabulate key studies

Author Response

Thank you very much for your critical but kind comments, which will enrich the manuscript, and I present my responses to your comments. Please see attached file. 

Reviewer 2 Report

The review manuscript submitted by Dubaniewicz summarizes potential link of mycobacterial heat shock proteins with sarcoidosis. This is an interesting topic as the pathogenesis of sarcoidosis remains elusive. Some but not all patients of sarcoidosis have a link of mycobacteria. However, in the second section, the review simply listed the data of three papers from the author’s group without proper digest. The third section was not clearly laid out. Providing subtitles might help. Additionally, the manuscript needs extensive editing.

Major comments

1.      L30. Could not find Figure 1 in the manuscript.

2.     Page 2-3. The observations summarized from ref 14-16 will be better if organized in a table or tables.

3.     L140-141. This section (P3-6) is badly organized. Subtitles should be provided to clarify the content of this section. For an example, NO was discussed in P4 and resumed in P6.

4.     The manuscript needs to be more carefully edited.  Few suggestions are listed in minor comments.

Minor comments

1.     Use capital letters for the abbreviation of heat shock protein (HSP) throughout the text, including HSP70, etc.  

2.     L48. Change “expressed” to “delivered”, as APC cells cannot express Mtb proteins.

3.     L66. Remove “rest of the”.

4.     L67-68. Change “the Mtb-hsp70, Mtb-hsp65, and Mtb-hsp16 expression” to “the presence of Mtb-HSP70, Mtb-HSPp65, and Mtb-HSP16”.

5.     L83, 85, and 87. Change “expression” to “level”. Be careful using “expression” throughout this paper

6.     L89-102. Rewrite this paragraph, which is lack of clarity. Such as “more often”, “higher percentages of …”. Are the controls in L97 positive or negative of the Ab? Change CIs in L102 and later to “ICs”.

7.     L101 and after change “immune complexes” and “CIs” to “ICs)

8.     L142. “Authors” – Specify.

9.     L143. What are the Mtb-HSPs? Individual protein or protein mixture?

10. L144. Specify “the same test groups”.

11. L145-147. Re-organize this sentence.

12. L153. Add period after SA.

13. L153-154 Check this sentence.

14. Change “hsp 16” to “HSP16” and “hsp 65” to “HSP65” in multiple places.

15. L170-171. “gen can responsible for reduced apoptosis in TB.”??

16. L172. Which strain is “the strain”?

17. L174. Intracellular bacteria – Specify whether the bacteria are restricted to mycobacteria (it seems not).

18.  L189. What does it mean of “in the present study?”

19. L197. The authors?

20. L204.  Change “in TB compared to SA and controls” to “in TB group compared to those in SA and control groups.”

21. L224-226. Re-organize this sentence.

22. L319. “(Human Leucocytic Antigens -HLA)” – HLA has been mentioned multiple times in earlier section.

23. L328-333. What the findings of this paragraph suggest? Do those bacterial species also linked to SA? What is the consequence of BCG vaccination on SA?

24. L334-363. Most of the contents in these paragraphs are beyond the scope of this review.

25. L334. “can also be activated”- should be “can also activate?”

26. A summary paragraph should be added in the end of the text.

Author Response

(The authors gave the same response as above.)

Reviewer 3 Report

Heat shock proteins are expressed in all pro- and eukaryotic organisms during stress conditions like phagocytosis, hypoxia, heat shock and oxidative stress. Mycobacterial hsp70, hsp65, and hsp16 may provide a link between latent infection and autoimmunity. Dubaniewicz group revealed that not whole mycobacterial but Mtb-hsp70, Mtb-hsp65, Mtb-hsp16 were detected in lymph nodes, sera and precipitated immune complexes in patients with SA and TB. The review summarized the systematic and elegant research works in Dubaniewicz group around heat shock proteins.

The review is well-prepared. My suggestion is that including some Schemes to illustrate each part of the review, such as: 1) How Heat shock proteins work as ‘danger signals’ in host and bacteria; 2) Mycobacterium tuberculosis complex and mycobacterial heat shock proteins in lymph nodes and sera of the patient with sarcoidosis and tuberculosis; et al.

Author Response

(The authors gave the same response as above.)

Round 2

Reviewer 2 Report

The manuscript has been well revised by the author. Providing of the figures and section titles significantly improved the quality of the manuscript. Although the author mentioned in the response that the text had been edited by a native speaker, edits are still needed (most of them are simple issues), except the journal will provide an editing service. Examples picked from Introduction were listed in Comments 1-8.

Comments

1.     L30 and 45.  Change “sarcoidosis” to “SA”.

2.     L37 and 45. Change “tuberculosis” to “TB”.

3.     L39. Change “material” to “materials”.

4.     L40. Change “blood release test” to “blood release assay”.

5.     Legend of Figure 1. Remove “s” from “antigen(s)”, “antigen (typo antygen) presenting cells”, “pattern recognition receptors”, “PAMPs – pathogen-associated molecular patterns”. At lease keep everything consistent.

6.     L46. Change “mycobacterium” to “M.”.

7.     L45-48. M. tuberculosis is an infectious agent. However, these proteins themselves are not infectious agents.

8.     L73. Should “primers” be “DNA amplified with primers”?

9.     Figure 2. Legend is not clear. What do the percentages (% of positive or levels?)­­, ODs, and “in INT/mm2” stand for? How many samples of each group were analyzed? Should they add the error bars and statistical analyses? If these data were published earlier, original reference should be cited in the legend.

10. Figure 2. The circulating Ab levels of Mtb-HSPs exhibited different patterns compared to HSPs detected in ICs. Each observation was described in text. However, the biological impact of the difference was not further discussed.

Author Response

Dear Reviewer,

please see the attachment  the file "Answers to Reviewer 2"

With best regards

Anna Dubaniewicz
